# Toll-like Receptor 4, Osteoblasts and Leukemogenesis; the Lesson from Acute Myeloid Leukemia

**DOI:** 10.3390/molecules27030735

**Published:** 2022-01-23

**Authors:** Øystein Bruserud, Håkon Reikvam, Annette Katharina Brenner

**Affiliations:** 1Department of Clinical Science, University of Bergen, 5020 Bergen, Norway; hakon.reikvam@uib.no (H.R.); annette.katharina.brenner@helse-bergen.no (A.K.B.); 2Department of Medicine, Haukeland University Hospital, 5021 Bergen, Norway

**Keywords:** toll-like receptor 4, osteoblast, mesenchymal stem cells, adipocyte, bone marrow, inflammation, acute myeloid leukemia, chemotherapy, allogeneic stem cell transplantation

## Abstract

Toll-like receptor 4 (TLR4) is a pattern-recognizing receptor that can bind exogenous and endogenous ligands. It is expressed by acute myeloid leukemia (AML) cells, several bone marrow stromal cells, and nonleukemic cells involved in inflammation. TLR4 can bind a wide range of endogenous ligands that are present in the bone marrow microenvironment. Furthermore, the TLR4-expressing nonleukemic bone marrow cells include various mesenchymal cells, endothelial cells, differentiated myeloid cells, and inflammatory/immunocompetent cells. Osteoblasts are important stem cell supporting cells localized to the stem cell niches, and they support the proliferation and survival of primary AML cells. These supporting effects are mediated by the bidirectional crosstalk between AML cells and supportive osteoblasts through the local cytokine network. Finally, TLR4 is also important for the defense against complicating infections in neutropenic patients, and it seems to be involved in the regulation of inflammatory and immunological reactions in patients treated with allogeneic stem cell transplantation. Thus, TLR4 has direct effects on primary AML cells, and it has indirect effects on the leukemic cells through modulation of their supporting neighboring bone marrow stromal cells (i.e., modulation of stem cell niches, regulation of angiogenesis). Furthermore, in allotransplant recipients TLR4 can modulate inflammatory and potentially antileukemic immune reactivity. The use of TLR4 targeting as an antileukemic treatment will therefore depend both on the biology of the AML cells, the biological context of the AML cells, aging effects reflected both in the AML and the stromal cells and the additional antileukemic treatment combined with HSP90 inhibition.

## 1. Introduction 

Bone marrow stromal cells constitute an interacting hematopoiesis-supporting network. Several of these cells are also important members of the stem cell niches [1,2]. The stromal cells include mesenchymal stem cells that can differentiate into osteoblasts and adipocytes [2] together with endothelial and perivascular cells, the sympathetic innervation, various immunocompetent cells and megakaryocytes that support other differentiation directions of normal hematopoiesis [1,2]. Hematopoietic cells are supported through cell-cell contact, soluble mediator release, and formation of the extracellular matrix [1,2]. Finally, osteoblasts [3], mesenchymal stem cells [4,5] and endothelial cells [6] even support leukemic hematopoiesis, including the development of acute myeloid leukemia (AML).

Osteoblasts synthesize almost all constituents of the bone matrix and regulate bone mineralization [7]. Toll-like receptor 4 (TLR4) is expressed by osteoblasts and is thereby a regulator of bone formation [8]. However, TLR4 is also expressed by primary human AML cells as well as other AML supporting bone marrow stromal cells and not only osteoblasts. Furthermore, the TLR4/Myeloid differentiation primary response 88 (MyD88) signaling in the leukemic cells seems important for leukemia cell growth, disease development and chemosensitivity in human AML. In this review we will therefore review and discuss TLR4 expression by and function in AML cells and bone marrow stromal cells, especially osteoblasts. We will also describe the role of TLR4 in the bidirectional crosstalk between AML cells and osteoblasts, and the possible use of TLR4 targeting therapy in AML therapy.

## 2. The Structure, Ligand Binding and Downstream Signaling of TLR4

### 2.1. TLR4 Expression by Bone Marrow Cells

TLRs are a group of interacting receptors that overlap with regard to ligand binding and downstream intracellular signaling. Several of the downstream mediators (e.g., Nuclear factor kappa-light-chain-enhancer of activated B cells/NFκB) are also targeted by other signaling pathways. The structure and function of TLR4 has been reviewed previously, and for this reason we only include a brief summary of its expression, namely ligand binding and signaling (Figure 1). TLR4 is expressed by hematopoietic stem cells as well as myeloid progenitors and mature neutrophils/monocytes/macrophages [9,10,11]. It is involved in the development of lymphoid cells [12], and it is expressed by various stromal cells [8]. It is also expressed by AML [13,14] and preleukemic myelodysplastic syndrome (MDS) cells [15]. Thus, TLR4 targeting will probably have both direct and indirect effects on leukemic hematopoiesis [8,14,16].

### 2.2. TLR4 Functions as a Promiscuous Receptor in the Bone Marrow Microenvironment

TLR4 binds proteins that express diverse pathogen-associated molecular patterns, the best characterized being lipopolysaccharide (LPS) [8]. However, endogenous agonists/ligands (Appendix A) (see Appendix A) are more relevant for the effects of TLR4 in the regulation of hematopoiesis. All of the listed agonists are present in the bone marrow microenvironment; they can either be released by bone marrow stromal or hematopoietic cells or are known regulators of normal hematopoiesis (Appendix A). Thus, all of these ligands would therefore be expected to act as TLR4 agonists and directly or indirectly influence the function of bone marrow stromal cells (including osteoblasts) as well as normal and leukemic hematopoiesis. TLR4 will thereby function as a highly promiscuous receptor also in the bone marrow microenvironment.

### 2.3. TLR4 Interacting Proteins; the Receptor Complex and the Downstream Signaling

The ligation of the receptor and the downstream signaling is summarized in Appendix A. Briefly, the TLR4 interacts with ligands (Section 3.2), co-receptors and adaptor proteins that determine the downstream signaling [8,17,18,19]. TLR4 ligation can activate additional downstream pathways through their crosstalk with the main MyD88-NFκB and TRIF(toll-interleukin 1 receptor-domain-containing adapter inducing interferon-β)-Interferon (IFN) Type1 pathways (Figure 1, Appendix A) (see Appendix A). However, a wide range of other proteins also interact with TLR4, and detailed descriptions of these proteins have been given in several recent reviews [17,18,19]. Appendix A includes a list of important interacting proteins, and they can be classified as (i) endogenous ligands; (ii) co-receptors or accessory molecules help in ligand recognizing; (iii) adaptor proteins initiating the downstream intracellular signal transduction; and (iv) negative regulators interact with TLR4 directly or indirectly and inhibit signaling [17,18,19].

The co-receptors bind ligands and transfer them to the TLR4/myeloid differentiation factor 2(MD-2) complex in the cell membrane. The downstream intracellular signaling depends on the recruitment of adaptor protein to this complex; MyD88 recruitment results in early NFκB activation whereas the TRIF recruitment seen after endosomal internalization of the TLR4 receptor complex causes activation of an alternative signaling cascade and finally results in an interferon type 1 response together with late NFκB activation [17,18,19,20]. Finally, TLR4 polymorphisms in the protein-encoding gene (rs1057317) seem to modulate TLR4 effects [21], whereas another polymorphism is located in an untranslated micro-RNA binding site and seems to be important for micro-RNA mediated regulation of TLR4 expression in osteoblasts [22].

## 3. TLR4 in Acute Myeloid Leukemia

### 3.1. Primary AML Cells Express Functional TLRs; the TLR4 Cytokine Responses Are Associated with NPM1 Mutations Whereas the Weaker TLR1/2 Responses Are Associated with Adverse Prognosis

The study by Brenner et al. [23] investigated the effect of TLR4 ligation on the constitutive cytokine release by primary human AML cells derived from 81 unselected patients (i.e., the use of increased cytokine as the readout will probably reflect mainly the downstream MyD88-NFκB signaling (Figure 1)) [24]. Previous studies had demonstrated that primary AML cells show constitutive release of a wide range of soluble mediators (i.e., interleukins, CCL(C-C motif ligand)/CXCL(C-X-C motif ligand) chemokines, growth factors, immunoregulators, proteases, and protease inhibitors), and a generally strong in vitro constitutive cytokine response has been associated with increased survival for younger AML patients receiving intensive chemotherapy possibly combined with allogeneic stem cell transplantation [25,26]. This TLR study showed that AML cell expression of functional TLR1/2 and TLR4 could be detected for most patients. Ligation of TLR4 caused a significant upregulation of the AML cell mediator release, including increased levels of interleukins (especially IL6 but also IL1β and IL1 receptor antagonist), chemokines (CCL2-5, CXCL1/5/8/10), growth factors (G-CSF/granulocyte colony-stimulating factor; GM-CSF/granulocyte-macrophage colony-stimulating factor, Hepatocyte growth factor) and proteases (Matrix metalloprotease(MMP)1/2/9). A generally strong TLR4 cytokine/mediator response was more common for the patient subset that already showed a strong constitutive cytokine release. Furthermore, the study identified a patient subset where increased mediator levels were observed in response not exclusively to TLR4 but also for TLR5 and with weaker increasing effects even for TLR1/2 and TLR7/8 [23]. A strong effect on the soluble mediator profile was associated with *NPM1* (Nuclophosmin-1) insertions, but not with any of the other generally accepted prognostic parameters (i.e., karyotype, FMS-related receptor tyrosine kinase 3 ligand (*FLT3*) mutations). The strong TLR responses were also associated with high mRNA expression of TLR receptors and proteins involved in TLR signaling (including TLR agonists), whereas patients characterized by generally weaker TLR responses showed higher expression of transcriptional regulators. Finally, less than half of the 81 consecutive patients were relatively young and received the most intensive treatment. For these patients, the strong TLR4 responses showed no independent association with prognosis whereas variations between patients in the generally weaker TLR1/2 responses showed a significant association with favorable prognosis that was independent of karyotype, NPM1 insertion, FLT3 internal tandem duplication, and etiology (i.e., secondary AML versus de novo).

To summarize, taken together the observations described above [23,24,25,26] suggest that the final effect of TLR ligation differs between patients; this difference seems to depend on differences in downstream signaling (i.e., high intracellular mediator levels/high extracellular mediator secretion) and transcriptional regulation (i.e., different NFκB context) between patients. Strong TLR4 responses are seen for most patients. In contrast, TLR1/2 responses are generally weaker, variation in TLR1/2 responses is generally wider between patients, and weak/no responses are associated with adverse prognosis.

### 3.2. TLR4 Expression and Function in AML

The study described in Section 3.1 had a functional readout (i.e., effect of receptor ligation) for TLR expression. Other studies have investigated TLR4 mRNA or protein expression:One study investigated mRNA expression of TLR2 and TLR4 in AML cells derived from 103 patients [27]. High expression of TLR2 and TLR4 was associated with decreased survival after intensive antileukemic therapy, and the association was stronger for TLR2 than for TLR4. This is similar to the previous functional study [23] that also described a relatively stronger prognostic impact of TLR1/2 responses.The prognostic impact of the TLR-associated cytokine response was also supported by a smaller study only investigating interleukin 6 (IL6) expression [14], and additional experimental studies in AML cell lines suggested that TLR4 inhibition had antiproliferative and proapoptotic effects mediated through upregulation of p21, TP53 (tumor protein p53, and BAD (BCL2 associated agonist of cell death).TLR4 expression is regulated at the transcriptional level by a suppressive effect of the Interferon regulatory factor 1 (IRF1), and studies in AML cell lines suggest that this factor also suppresses expression of the adhesion molecule ICAM1 (intercellular adhesion molecule 1)and in addition has an antiapoptotic effect [28].TLR4 as well as IL6 and TNFα (tumor necrosis factor α) expression can be increased by in vitro induction of a dendritic AML cell phenotype in primary leukemic blasts after exposure to GM-CSF, IL4, and TNFα [29].Studies in AML cell lines suggest that TLR4-NFκB signaling causes resistance to the antileukemic agent fludarabine, possibly by counteracting the fludarabine-induced increase of the tumor suppressor thioredoxin-interacting protein [30].The S100A8 and S100A9 proteins are important for AML cell proliferation, cell cycle regulation, differentiation, and chemosensitivity [31,32]. High S100A8 or a high S100A8/S100A9 ratio is associated with an adverse prognosis and chemoresistance in AML [33,34]. S100A9 binds TLR4 and thereby activates p38, Extracellular signal-regulated kinase 1(ERK)1/2 and Jun N-terminal kinase signaling, and this TLR4-associated S100A9 effect seems to be a part of the mechanism behind the adverse prognostic impact of S100A8.

Taken together these studies suggest that TLR4 expression by AML cells is associated with an adverse prognosis (i.e., a part of a high-risk leukemic cell phenotype), and TLR4 inhibition can have a chemosensitizing and/or direct antileukemic effect. However, it should be emphasized that many of these observations are based on studies in AML cell lines, and further experimental as well as clinical studies are needed especially to (i) clarify the molecular effects in primary AML cells, (ii) the possibility of patient heterogeneity with regard to the antileukemic effects of TLR4 targeting, and (iii) whether TLR4 expression is an independent prognostic marker when adjusting for generally accepted prognostic parameters in AML.

### 3.3. The Crosstalk between Primary AML Cells and Osteoblasts

The crosstalk mediated by the local cytokine network between primary AML cells and osteoblasts was investigated in a previous study [3]. These studies were based on examination of both an osteoblastic cell line and normal osteoblasts. The cross-talk between primary AML cells and osteoblasts mediated by modulation of the local cytokine network caused increased AML cell proliferation, including increased proliferation of clonogenic progenitors, but did not influence spontaneous in vitro apoptosis. Both IL1β and GM-CSF were involved in this growth-enhancing cross-talk. Increased levels of the proangiogenic mediator IL8/CXCL8 were also observed in these cocultures, suggesting that that the crosstalk will have an additional an indirect AML-supporting effect through stimulation of local angiogenesis (see Section 6).

In a recent proteomic study, we investigated the effect of the constitutive AML cell release of soluble mediators on the proteomic profile of normal mesenchymal stem cells (MSCs) [35]. We observed that the AML effect on the global MSC proteomic profiles differed between patients, and we detected two main AML patient subsets with regard to the AML effect on normal MSCs. There were no detectable proteomic signs of altered or induced MSC differentiation towards osteoblasts or adipocytes for any of the two subsets. However, AML cells increased the expression of TLR2 and IRAK3 (interleukin 1 receptor associated kinase 3), the last mediator being important for intracellular TLR signal transduction. Thus, primary AML cells modulate the TLR expression profile and downstream TLR signaling in MSCs. However, additional studies are needed to clarify whether these effects will influence the capacity of MSC differentiation.

## 4. TLR4 in MDS

Myelodysplastic syndromes are regarded as a group of preleukemic diseases, and TLR4 seems to be involved in the pathogenesis also for this disease [15,36,37]:TLR4 and TLR2 are expressed by almost all myeloid bone marrow lineages and also B lymphocytes in MDS; the level is upregulated compared to healthy controls.Anti-TNFα antibodies reduce both the constitutive TLR4 expression and the LPS-induced TLR4 increase of MDS bone marrow cells.TLR4 expression correlates with increased bone marrow apoptosis, it is mainly detected in apoptotic cells and in CD(cell differentiation)34^+^ cells, and lipopolysaccharide (LPS) seems to further increase the apoptosis. Thus, surprisingly TLR4 may have a proapoptotic effect in MDS that possibly contributes to cytopenia, and the apoptotic MDS cells release the TLR4 ligand HMGB1 (high mobility group box 1).The TLR4 expression is not altered by progression to AML.

The progression from preleukemic MDS to AML illustrates that leukemogenesis in AML is a stepwise process. TLR4 expression is similar both at the MDS step and after progression to AML, but despite these similarities the functional impact of TLR4 seems to differ between steps with a proapoptotic effect only in the preleukemic malignant cells.

## 5. Effects of TLR4 on Osteoblast Differentiation

### 5.1. Osteoblastic Differentiation from MSCs

The differentiation of osteoblasts from MSC depends on the balance between adipogenic and osteoblastic differentiation, a process influenced by several factors including [38,39]:Transcriptional regulation. The Runx2 (RUNX family transcription factor 2) transcription factor is important for the initiation of osteoblastic differentiation together with the zinc finger transcription factor osterix and the SOX9 (SRY-box transcription factor 9) transcription factor. Runx2 is also an inhibitor of adipocyte differentiation. Furthermore, important transcriptional regulators that stimulate adipogenic differentiation are Peroxisome proliferation-activated receptor γ (PPARγ) and CCAAT-enhancer binding protein α (CEBPα), whereas other transcription factors seem to inhibit adipogenic differentiation (including GATA binding protein 2/GATA2, Forkhead transcription factor 1, Homeobox C8). Thus, osteoblastic MSC differentiation depends on the balance between these transcriptional regulators.Signaling pathways. Several of the bone marrow morphogenic proteins can stimulate both osteoblastic and adipogenic differentiation, possibly depending on the biological context [38,39]. In contrast Wnt (Wingless and Int-1)/β-catenin signaling has proosteoblastic and antiadipocytic effects; these effects can be antagonized by sclerosin. Studies of gravity effects on bone formation suggest that integrin, mitogen-activated protein (MAP) kinase, RhoA (Ras homolog family member A) activity, ERK1/2 and p38 signaling are all involved in MSC/osteoblastic differentiation through effects on Runx2 and/or PPARγ. Finally, notch signaling also seems involved in osteoblastic differentiation. Thus, osteoblastic differentiation involves complex interactions between several intracellular signaling pathways.Micro RNA. Several micro RNAs are involved in osteoblastic differentiation and most of them showing antiosteoblastic and proadipocytic effects; a common mechanism for several of them seems to be Runx2/osterix modulation.Aging. Bone marrow aging is characterized by increased adipogenesis [1,40].The final fate of osteoblasts is to become either bone lining cells or an osteocyte, or to undergo apoptosis [39].

The effect of TLR4 ligation on MSC differentiation seems to differ between MSC subsets. For example, TLR4 ligation can induce osteoblastic differentiation of umbilical cord MSCs whereas it inhibits the differentiation of bone marrow MSCs at least in certain experimental models [41]. The umbilical cord MSCs also seem to have a more restricted cytokine response after TLR4 ligation. Studies in a murine MSC cell line also suggest that TLR4 ligation inhibits osteoblastic differentiation [42]. In our opinion the only possible conclusion is that MSC differentiation to osteoblasts is regulated by complex mechanisms at the transcriptional level and by several intracellular signaling pathways; the effect of TLR4 ligation on these complex mechanisms is difficult to predict and possibly dependent on the biological context.

### 5.2. Effects of TLR4 on Osteoblast Differentiation

Several studies have investigated the effect of TLR4 ligation on osteoblasts [20,43,44,45]. TLR4 ligation has various effects and usually inhibitory effects on osteoblastic differentiation:MSCs express several functional TLRs including TLR4. At least in certain experimental models, TLR4 ligation did not influence proliferation, survival, or immunosuppressive ability of MSCs [43].TLR4 ligation can inhibit both adipogenic and osteoblastic MSC differentiation. TLR4 ligation also suppressed BMP-2 induced differentiation through an NFκB dependent effect. Furthermore, TLR4 ligation decreased the expression of certain stem cell markers by umbilical cord MSCs, but this was not associated with osteoblastic signs of differentiation [46]. A study of murine bone marrow MSCs could not detect any effect of TLR4 ligation of the stemness markers CD34, CD11b, CD44 or Sca-1 either [43].TLR4 ligation suppresses expression of osteocalcin, osteopontin and several osteoblast-associated matrix proteins, but the mRNA expression of CD14 and TLR4 is increased.TLR4 ligation can protect MSCs from oxidative-stress induced apoptosis and in this model MSC proliferation was increased through a PI3K (phosphoinositide 3-kinase)/Akt (AKT serine/threonine kinase 1) mediated effect [47].

However, two recent studies suggest that the effect of TLR4 on osteoblast differentiation is more complex. First, a study of murine bone marrow MSCs cultured in vitro in a medium stimulating osteogenic differentiation showed that TLR4 ligation promoted proliferation and osteogenic differentiation together with increased IL1β/IL6 release [48]. These effects were dependent on Wnt signaling; Wnt3a was important for the effect on proliferation whereas Wnt5a was essential for induction of osteogenic differentiation. Second, a study of human osteoblasts suggested that osteogenesis could be upregulated/ enhanced only when cells were stimulated with weak TLR4 agonists [49]. Thus, TLR4 stimulated osteoblastic differentiation is seen in certain experimental models (i.e., the biological context), and the final effect of TLR4 ligation may also depend on the nature of the TLR4 ligand.

The periodontal ligament is the connective tissue that links teeth and alveolar bone, and it contains multipotent stem cells that can differentiate into various mesenchymal cells, including osteoblasts [50,51]. Two studies have investigated the effect of TLR4 on osteoblastic differentiation of such stem cells. First, the periodontal cells were cultured in medium only supplemented with fetal bovine serum. Inhibition of TLR4 then increased osteoblastic differentiation as well as migration and proliferation of the cells, observations suggesting that TLR4 ligation inhibits osteoblastic differentiation of these cells under these experimental conditions [50]. Second, in another study the culture medium was in addition supplemented with dexamethasone, glycerophosphate and ascorbic acid which will increase osteoblastic differentiation. TLR4 agonists had an antiproliferative effect and inhibited osteoblastic differentiation also in this model [51]. Inhibition of differentiation was associated with reduced levels of Runx2 and osterix. Thus, both studies suggest that TLR4 has an inhibitory effect on osteoblastic differentiation of these mesenchymal cells. In contrast, TLR4 seems to enhance osteoblastic differentiation of adipose-derived stem cells; this effect seems to be mediated by modulation of the micro-RNA network [52].

A previous study described that five different mesenchymal progenitor cell subsets could be derived from periodontal ligaments, and in a recent study the effect of TLR4 on osteoblastic differentiation of the CD105+ progenitor subset was investigated [53]. These authors describe that TLR4 ligation under osteogenic conditions increased Runx2 expression, alkaline phosphatase levels and mineralized matrix deposition by these CD105+ cells [53]. Furthermore, the role of TLR4 has also been investigated in the context of fracture healing during Severe acute respiratory syndrome coronavirus 2 (SARS-CoV-2) infection. In this context miR-4485 was upregulated and caused reduced TLR4 expression leading to reduced osteogenic differentiation [54]. Finally, TLR4 ligation can inhibit the further differentiation of osteoblasts into osteocytes [55].

### 5.3. The TLR4 Induced Increase in Extracellular Release of Soluble Mediators by Osteoblasts

The TLR4 induced increase in osteoblastic release of soluble mediators includes a wide range of mediators and especially proinflammatory mediators [8]. The following descriptions are based on the overall results from studies of human and animal cells:Interleukins. Proinflammatory interleukins are increased in response to TLR4 ligation, including IL1β [49], IL6 [53] and IL8/CXCL8 [48].Chemokines. Both CCL and CXCL chemokines are increased in response to TLR4 stimulation, including CCL2 [56], CXCL1 [56], CXCL8/IL8 and CXCL10 [57].Other cytokines. TLR ligation increases the release of TNFα [58] and Vascular endothelial growth factor (VEGF) [59].Proteases. TLR4 ligation increases matrix metalloprotease 13 (MMP13) [60].

It should be emphasized that osteoblasts release a wide range of soluble mediators. A recent study compared the proteomic profile of supernatants derived from cultures of human osteoblasts and MSCs [61]. A total of 1379 molecules were quantified to be released by these cells, and as many as 340 proteins belonged to the Gene Ontology term “Extracellular matrix”. Both cell types released a wide range of functionally heterogeneous proteins including extracellular matrix molecules (especially collagens), proteases, cytokines and soluble adhesion molecules, but also intracellular molecules including chaperones, cytoplasmic mediators, histones and non-histone nuclear molecules. Few differences in protein concentrations were observed: 82 proteins were more abundant in MSCs and 36 proteins were more abundant for osteoblasts. To the best of our knowledge the effect of TLR4 on the overall proteomic profile of constitutively released extracellular proteins has not been investigated. Thus, only a small minority of proteins constitutively released by osteoblasts and MSCs have been investigated with regard to the possible influence of TLR4/TLR4 inhibition.

The experience from other cell types suggest that the expression of a wide range of cytokines is controlled by TLR4 and/or NFκB, including several interleukins as well as CCL and CXCL chemokines, and this is true both for MSCs [5] and monocytes [62]. Future studies should therefore clarify whether TLR4 ligation has broader effects on osteoblastic release of soluble mediators than the relatively few mediators that have been investigated until now.

### 5.4. Effects of TLR4 on Proliferation and Survival of Osteoblasts

Several studies have investigated the effects of TLR4 on osteoblast proliferation. First, the TLR4 ligand HMGB1 did not alter osteoblast proliferation [63] or viability [53]. Second, a study of human periodontal ligament MSCs suggested that TLR4 could inhibit osteoblastic differentiation and at the same time inhibit proliferation and migration [50]. Finally, a study of the bone marrow stromal cell line HS-5 described a proapoptotic effect of HSP60 mediated by TLR2/TLR4/NFκB [64]. This last study may not be representative for human osteoblasts, but the first two studies suggest that the effect of TLR4 on mesenchymal or osteoblastic cells depends on the source of the cells and/or the experimental conditions.

### 5.5. Other TLR4 Effects on Osteoblasts

Studies in cell lines suggest that TLR4 ligation induces expression of nitric oxide synthase and thereby the production of nitric oxide [65]. Endogenous or exogenous nitric oxide has toxic effects and can induce apoptosis through mitochondria-dependent mechanisms [66], but whether nitric oxide has effects on the regulation of survival/apoptosis in osteoblasts or their neighboring cells in the bone marrow is not known. Furthermore, TLR4 ligation by HMGB1 inhibits the migration of osteoblasts without having any effect on proliferation; this effect on migration seems to be NFκB dependent [63]. Another study also described decreased migration by TLR4 ligation [50]. Finally, TLR4 agonists can increase the expression of cyclooxygenase 2 and prostaglandin E synthase and thereby increase prostaglandin E_2_ production in murine osteoblasts [67].

### 5.6. Effects of TLR4 on Other Hematopoiesis-Supporting Bone Marrow Stromal Cells

The bone marrow has several stromal cells that influence both normal and leukemic hematopoiesis, and several of these cells are also important for the formation of the various stem cell niches in the bone marrow. The influence of TLR4 on important bone marrow stromal cells is summarized in Table 1 [1,58,68,69,70,71,72,73,74,75]. All of these cells can support normal hematopoiesis and normal hematopoietic stem cells [1], many of them are also important in AML leukemogenesis and the use of TLR4 inhibition in AML therapy will thus have both direct effects on the leukemic cells (Section 3) and indirect effects mediated through the various leukemia-supporting nonleukemic stromal cells.

### 5.7. MSCs and Osteoblasts Express a Wide Range of TLRs

MSCs can express TLR1-9 [58,68]. Many of these receptors show downstream signaling through the same intracellular mediators and may thereby interfere with TLR4 signaling and TLR4 effects on MSC differentiation, e.g., TLR2 ligation shows a MyD88-dependent inhibition of osteoblastic differentiation by MSCs [68]. This example clearly illustrates that the effect of TLR4 ligation on osteoblastic differentiation is influenced by the biological context, i.e., in this example signaling by other TLR receptors. However, the results support the hypothesis that TLR-initiated signaling through MyD88 initiated by various TLRs has an inhibitory effect on osteoblastic MSC differentiation.

### 5.8. Concluding Comments: The Effects of TLR4 on MSC and Osteoblast Differentiation

TLR4 has very complex effects on osteoblasts. The results from studies of osteoblastic differentiation of MSCs seem to be conflicting, but the likely explanation for these differences is probably that the TLR4 effect on MSC/osteoblast differentiation depends on the biological context. This is also supported by in vivo studies in various animal models showing that TLR4-mediated effects of diet on osteoblastic differentiation differ between mouse strains [76]. Some of the observations may also be relevant only for the animal models or cell lines investigated. Aging effects may also contribute to the variation in TLR4 effects together with the type of TLR4 ligand or the balance between various TLR ligands (discussed above). Furthermore, the balance between various TLR4 ligands in the bone marrow microenvironment may also be important, possibly together with TLR4 genetic polymorphisms. However, it should be emphasized that despite the several factors that influence and modify the TLR4 effects on osteoblasts, the cytokine response with increased release of cytokines/chemokines/growth factors has been described in a wide range of experimental models.

Previous studies have shown that fibroblasts and monocytes derived from AML patients have altered expression of several genes involved in hematopoietic stem cell control, and these effects alter their capacity of supporting hematopoiesis [77]. It is not known whether similar effects are seen for osteoblasts derived from AML patients.

## 6. TLR4, Osteoblasts and Angiogenesis

Increased bone marrow angiogenesis is important for the development of AML, and it is associated with chemoresistance [78,79]. Osteoblasts release a wide range of angioregulatory cytokines, including both CCL and CXCL chemokines, and recent studies have investigated the role of osteoblasts in angiogenesis [75,80,81,82,83,84]. These investigators mainly used experimental models based on coculture of human osteoblasts and circulating endothelial cells with formation of perfused vascular structures. TLR4 ligation increased both angiogenesis and osteogenesis in such cocultures. These effects were associated with upregulation of TLR4 together with MyD88 and osteogenic markers in the osteoblasts. A similar upregulation of TLR4 was also seen for the endothelial cells together with increased proliferation, altered migration and actin filament reorganization. Furthermore, TLR4 ligation increased the levels of the angioregulatory cytokines angiopoietin-1, angiopoietin-2 and VEGF together with the expression of the adhesion molecules ICAM1 and E-selectin. Finally, VEGF is also releases by osteoblasts, and this release is mediated by TLR4-PI3K-dependent signaling [59]. Taken together these observations suggest that TLR4 ligation mediates a proangiogenic effect in the crosstalk between osteoblasts and endothelial cells, and this effect may then be important for leukemogenesis in human AML [78,79].

## 7. TLR4 Mediated Immunoregulation in AML; the Importance of Genetic Polymorphisms

### 7.1. TLR4 Mediated Regulation of Inflammation in Patients Receiving Intensive Chemotherapy

TLR4 genetic polymorphisms may influence immunoregulation and thereby be important in aging and/or development of proinflammatory diseases in the elderly [85]. Previous studies have therefore investigated the possible influence of TLR4 polymorphisms on the risk of infections in AML patients. One study investigated the bone marrow TLR4 expression at the time of AML diagnosis before start of intensive antileukemic therapy and during treatment-induced bone marrow failure after induction therapy [86]. The TLR4 expression by immunocompetent cells was significantly higher for patients developing sepsis during the period of treatment-induced neutropenia, and the levels were also higher in bacterial compared to fungal infections. These observations suggest that the ability to increase TLR4 expression during inflammation is maintained in these severely immunocompromised patients. Other studies have investigated the possible importance of TLR4 genetic polymorphisms and the clinical course for acute leukemia patients receiving intensive and potentially curative acute leukemia treatment:One study investigated eight single nucleotide polymorphisms (SNPs) in 194 children with acute lymphoblastic leukemia (ALL), and the four SNPs rs10759931, rs11536889, rs1927911 and rs6478317 were associated with increased risk of treatment-induced cytopenia [87].The TLR4 Asp299Gly polymorphism seems to influence the risk of sepsis after AML therapy [88]. Another study also described an association between this SNP and the risk of both sepsis and pneumonia; the same associations were described for the Thr399Ile TLR4 polymorphism whereas the TLR2 polymorphism Arg753Gln was only associated with pneumonia [89].With regard to invasive fungal infection one study described an association with Dectin-1 SNPs, whereas they observed no association with TLR4 and MyD88 SNPs [90]. On the other hand, another study described an association between invasive aspergillosis and the Asp299Gly/Thr399Ile variants in the donor or recipient after allogeneic stem cell transplantation [91].The TLR4 Asp299Gly polymorphism was associated with relapse risk in one previous study [88], and a study of European children with ALL suggested an association between TLR6 SNPs and risk of leukemia [92].

Thus, these studies suggest that TLR4 SNPs influence hematopoietic reconstitution after chemotherapy [87]. TLR4 polymorphisms together with TLR2/Dectin-1 SNPs may also be important for the function of remaining immunocompetent cells during severe chemotherapy-induced neutropenia [88,89,90,91]. TLR SNPs may even influence the risk of leukemia [88,92]. However, it should be emphasized that it is not known whether these effects are mediated via bone marrow stromal cells (e.g., by osteoblasts or MSCs) or represent direct effects on hematopoietic cells [3].

### 7.2. TLR4 Mediated Regulation of Inflammation in Allotransplant Recipients

Observations in animal studies suggest that TLRs are involved in the development of chemotherapy-induced mucositis, graft versus host reactions and graft versus leukemia reactivity in allogeneic hematopoietic stem cell transplantation [93]. This is also true for TLR4:A recent study of 816 patients concluded that a TLR4 haplotype including two SNPs in strong linkage disequilibrium (D299G and T399I) were associated not only with the risk of Aspergillus infections but also 3-years non-relapse mortality [94]. This association was only observed for patients with unrelated donors but not for patients with family donors, and this may be the reason why TLR4 associations were not observed in another study [95].Another study described a significant association between the cosegregating Asp299Gly/Thr399Ile TLR4 polymorphisms and fungal colonization, whereas the susceptibility to fungal infections (predominantly pneumonia) was significantly decreased in the presence of the same variants [96].The TLR4 SNPs Thr399Ile (rs4986791) or Asp299Gly (rs4986790) are associated with reduced responsiveness to endotoxins and were investigated in 166 allotransplanted children [97]. Asp299Gly was present in 13% of the patients and 14% of the donors whereas Thr399Ile was found in 13% of the patients and 15% of the donors. The incidence of hemorrhagic cystitis was significantly lower in patients with Asp299Gly (0% versus 23%; p = 0.009) and in patients who underwent transplantation from a donor with Asp299Gly (4% versus 23%; p = 0.05). Multivariate analysis revealed age, busulfan conditioning, and absence of Asp299Gly as independent risk factors for hemorrhagic cystitis.A small study including only 77 recipients suggested that the two TLR4 polymorphisms Asp299Gly and Thr399Ile were associated with the risk of Gram-negative bloodstream infection, but the association reached only borderline significance [98].TLR4 is important for maturation of dendritic cells and it promotes allogeneic T-cell proliferation and T-helper cell 1 development; TLR4 inhibition/deletion leads to increased levels of the Th2 cytokines IFN-γ and IL-10 and thereby protects from graft versus host disease (GVHD) [99]. Furthermore, both animal studies and studies of transplant recipients suggest that TLR4 expression/ligation is involved in the development of acute GVHD both in the skin, liver and gastrointestinal tract [99,100,101,102].

Thus, TLR4 seems to be involved in posttranspant immunoregulation and the development of acute GVHD, an important risk factor for of non-relapse mortality. This effect may also influence the risk of relapse, but it is not known whether TLR4 is important for the specific graft versus leukemia reactivity. These observations further support our conclusion from Section 7.1; TLR4 is important for the defense against infections in immunocompromised patients treated for hematological malignancies, but again it should be emphasized that it is not known whether these effects are mediated via the hematopoiesis/stem cell supporting stromal cells (e.g., osteoblasts/MSCs) or represent direct effects on the immunocompetent cells [3].

## 8. Pharmacological Targeting of TLR4; a Possible Therapeutic Strategy in Hematological Malignancies?

### 8.1. The Development of TLR4 Inhibitors

Several TLR4 inhibitors have been developed [18,103]; the targeting of TLR4 has then been based on various molecular strategies including (i) blocking of the binding between ligands and TLR4/TLR4-associated molecules (lipopolysaccharide binding protein/LBP or CD14 binding), (ii) blocking of the binding between TLR4 and its associated molecules (e.g., MD2-TLR4 binding), (iii) modulation of TLR4 expression, or (iv) targeting of intracellular signaling downstream to the receptor [103]. The inhibitors include both small molecules (i.e., molecular weight < 1 kDa) and monoclonal antibodies. Both statins and TNFα inhibitors can be used to modulate the expression of TLR4 [103,104,105,106], and repurposing of tricyclic antidepressants or opioid derivatives has also been suggested [18,103]. The statin effect on TLR4 expression has even been detected in myeloid leukemia cells [107].

TLR4 inhibitors are now in clinical trials [108,109,110,111]. Randomized studies of specific TLR4 inhibitors in patients with sepsis/severe infections have failed to demonstrate a convincing beneficial effect, but the inhibitors are also investigated in autoimmune inflammatory diseases. However, the clinical experience so far is that the adverse effects are few and the toxicity seems acceptable. Hematological toxicity is often dose-limiting for the present AML therapy but has not been a major problem in clinical studies of TLR4 inhibitors [108,109,110,111].

TLR4 agonists have been developed and are used as vaccine adjuvants [101]; whether TLR4 agonists should be tried to enhance antileukemic graft versus leukemia reactivity in allotransplant recipients require further investigation in experimental studies. An alternative strategy may be to use TLR4 agonists as cancer vaccine adjuvants.

### 8.2. The Question of Patient Heterogeneity and the Importance of Biomarker Identification

Several observations suggest that patients will differ with regard to the direct effects of TLR4 on AML cells, and may therefore be heterogeneous with regard to the antileukemic effect of TLR4 targeting:TLR4 polymorphisms will influence the signaling and function of TLR4 ligation (Section 2.3, Section 7.1 and Section 7.2).The levels of various S100A molecules differ between patients and TLR4 inhibition may then reduce the chemosensitivity associated with predominant S100A9 expression [31,32,33,34].The constitutive AML cell mediator release cytokine release profile and thereby the context of the TLR4 induced cytokine response will differ between patients [24,25,26]. The effect of TLR4 targeting on MSCs/osteoblasts seems to depend on the biological context (Section 5.1 and Section 5.2); due to the functional heterogeneity of AML cells with regard to their constitutive protein release the biological context/microenvironment of the osteoblasts and thereby the effect of TLR4 targeting will differ between AML patients [25,35,45,112,113,114,115,116]. However, although the heterogeneity between patients with regard to constitutive cytokine release by the AML cells may influence the MSC-osteoblast differentiation, the AML supporting MSC effect is observed independent of this patient heterogeneity [3].The functional status of downstream crosstalking pathways differs between patients and will therefore influence the effect of TLR4 inhibition [117,118,119]. Recent studies suggest that the crosstalk with IL1 signaling will be of particular importance. The downstream signaling of both the IL1 receptor and TLR4 is mediated through MyD88/IRAK1 [120,121,122], and recent studies suggest that signaling through this pathway in primary AML cells is important for leukemic cell growth and survival (i.e., leukemia progression) [121,123] and also for the AML-supporting effects in the stem cell niches that are formed by bone marrow stromal cells [124,125]. These effects can be mediated both by IL1 as well as other IL1 cytokine family members (e.g., IL33) [126,127]. The antileukemic effect of IRAK1 targeting does not show any association with the generally accepted prognostic factors associated with chemoresistance in human AML [123]. This importance of MyD88-IRAK1 in AML supports further investigation of TLR4/Myd88/IRAK1 targeting as a possible therapeutic strategy in human AML.

Thus, the question of patient heterogeneity with regard to the antileukemic effect of TLR4 inhibition should be carefully addressed in future clinical studies, and biomarkers that reflect the antileukemic effect of TLR4 inhibition should be identified.

## 9. Conclusions

TLR4 targeting is now tried in vaccination (agonists) and in the treatment of inflammatory diseases (antagonists). The toxicity of this therapeutic strategy seems to be acceptable [108,109,110,111]. Previous studies have described associations between both leukemic cell TLR4 expression/signaling as well as certain TLR4 single nucleotide polymorphisms and risk of relapse in human AML [23,27,31,32,33,34,88,92]. These observations suggest that therapeutic TLR4 targeting should be further investigated in this disease. TLR4 targeting would then be expected to have both direct effects on the leukemic cells as well as indirect effects mediated through bone marrow stromal cells, including osteoblasts that are important in the formation of stem cell niches. Furthermore, the differentiation and function of osteoblasts depend on the biological context and at the same time AML is a heterogeneous disease. This means that the biological context of osteoblasts depends on the AML cells and thereby differs between patients. Future clinical studies of TLR4 targeting in AML should therefore include detailed biological characterization of both bone marrow MSCs/osteoblasts and AML cells to further investigate (i) the molecular mechanisms responsible for the prognostic impact of TLR4; (ii) the molecular crosstalk between leukemic and stromal cells; and (iii) how these mechanisms differ between patients and whether TLR4 targeting will be effective only in certain subsets of AML patients [120]. This will be necessary to identify new biomarkers for susceptibility to TLR4 targeting strategies since the experience with MyD88/IRAK1 inhibition suggests that the antileukemic effect of this strategy is not associated with established prognostic factors for chemosensitivity in human AML [123]. Finally, the effect of TLR4 on immunocompetent cells has to be carefully considered if TLR4 targeting is used as a remission-maintaining strategy [128], especially if combined with allogeneic stem cell transplantation and/or other forms of antileukemic immunotherapy.

## Figures and Tables

**Figure 1 molecules-27-00735-f001:**
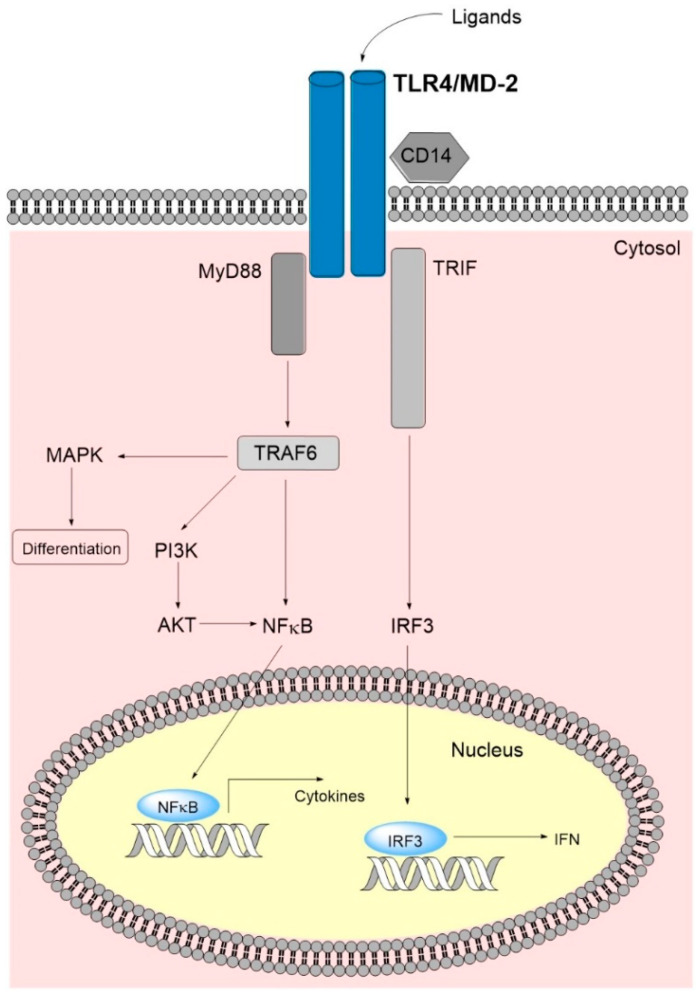
A simplified overview of the TLR4 main signaling pathway. Binding of ligands leads to TLR4-dimerization causing a conformational change in the C-terminal signaling domain and subsequent binding of adaptor molecules. The latter initiate signaling cascades via the MyD88 dependent and the MyD88 non-dependent/TRIF-dependent pathways. Ultimately, TLR4 signaling leads to nucleus internalization of NFκB and IRF3 and subsequent transcription of inflammatory cytokines and type 1 IFN. The MyD88 dependent pathway interrelates, amongst others, with the MAPK and PI3K/AKT pathways, where the latter additionally signals via NFκB, further increasing cytokine transcription. Abbreviations: AKT, AKT serine/threonine kinase 1 (protein kinase B); IFN, interferon; IRF3, interferon regulatory factor; MAPK, mitogen-activated protein kinase; MD-2, myeloid differentiation factor 2; MyD88, myeloid differentiation primary response gene 88; PI3K, phosphoinositide 3-kinase; TLR4, toll-like receptor 4; TRAF6, tumor necrosis factor receptor-associated factor 6; TRIF, toll-interleukin 1 receptor-domain-containing adapter inducing interferon-β.

**Table 1 molecules-27-00735-t001:** The importance of TLR4 for various non-leukemic bone marrow cells; a summary of TLR4 effects on the various stromal cells [1,58,68,69,70,71,72,73,74,75].

**Osteoclasts [58,70,71].** TLR4 ligation prolongs osteoclast survival and this effect is associated with activation of various downstream mediators (i.e., Akt, NFκB, ERK). It can also increase osteoclastogenesis. However, this effect depends on the presence of TNFSF11/RANKL (official name TNFSF11, TNF superfamily member 11), and in the absence of this mediator TLR4 has an inhibitory effect. Finally, osteoblasts can function as regulators of osteoclastogenesis, and both TLR4 and TLR9 ligation can modulate this osteoblast effect.
**Endothelial cells [69,72].** Endothelial cells express TLR4 together with TLR1, 2, 5, 6 and 9. TLRs are involved in (i) upregulation of microvascular endothelial cell expression of inflammatory mediators, (ii) modulation of endothelial cell permeability and (iii) modulation of the coagulation cascade.
**Pericytes [69,73,74,75].** TLR4 is expressed by pericytes together with TLR2. TLR4 ligation causes upregulation of TLR2 as well as TLR4, several proinflammatory interleukins together with CCL/CXCL chemokines, VCAM1 (Vascular cell adhesion molecule 1), ICAM1 and CD14. Finally, TLR4 inhibition leads to increased microvessel permeability.
**Adipocytes [69].** Adipocytes express several TLRs, including TLR1, 2, 4, 7 and 8. Both TLR2 and TLR4 ligation causes increased IL6/TNFα release as well as increased NFκB expression.
**Neutrophils [69].** Neutrophils express several TLRs, including TLR4 but also TLR1, TLR2 and TLR5-10. TLR ligation induces cytokine release and superoxide production.
**Monocytes/macrophages [62,69].** These cells express TLR1-10 and TLR13. TLR ligation causes cytokine release and increased phagocytosis. TLR4 ligation causes release of several interleukins, chemokines and growth factors.
**T cells [69].** Various T cell subsets differ in their TLR4 expression. The receptor is expressed by CD4^+^ but probably not by most CD8^+^ T cells; it is also expressed by γδ T cells and also by Treg cells.

## Data Availability

Not applicable.

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
