# Peer review of "Toll-like Receptor 4, Osteoblasts and Leukemogenesis; the Lesson from Acute Myeloid Leukemia"

_molecules, 2022, doi:10.3390/molecules27030735_

Round 1

Reviewer 1 Report

This review is very interesting, well writing and really rich in information. It addresses an emerging topic that involves a broad scope of basic research with promising translatal aspects. However, I would like to make two suggestions to the authors. First, in the introduction I would include the need for this review, that is, I would emphasize the existing gaps in the literature that the paper fills. Moreover, in the face of fairly extensive work, the conclusions are too limited. In my opinion, the foundations should be laid for a future research agenda focusing on several points.

Author Response

This review is very interesting, well writing and really rich in information. It addresses an emerging topic that involves a broad scope of basic research with promising translational aspects. However, I would like to make two suggestions to the authors. First, in the introduction I would include the need for this review, that is, I would emphasize the existing gaps in the literature that the paper fills. Moreover, in the face of fairly extensive work, the conclusions are too limited. In my opinion, the foundations should be laid for a future research agenda focusing on several points.

Response: We have added a brief comment to the Introduction section (Section 1). We have also rewritten the conclusion of our article (Section 9). We hope this is acceptable. To further support our conclusion we have also added a comment on the importance of Myd88/IRAK1 signaling in human AML together with nine new references that are used in chapter (section 8.2).

Reviewer 2 Report

Overall there are few references in table 2, and 89 is quite repeated.
Why did you describe TLR4 structure and function so in depth? It is known and has been reviewed quite a lot.
What do you mean by preleukemic MDS? Yes, we know that MDS can go to AML.
Overall I do not understand the reason for making this review. It is overly focused to the point that it seems like a research question.
I do not find a reason for this review to be published.

Author Response

Overall there are few references in table 2, and 89 is quite repeated.

Response: Reviewer 3 wanted us to reduce the overall number of references. The references included in the original Table 2 (Table 1 in the Revised Version) are sufficient to cover all cell types described in the table. For this reason we have not added new references in the table. We hope this can be accepted.

Why did you describe TLR4 structure and function so in depth? It is known and has been reviewed quite a lot.

Response: In our opinion it will be a help for readers who are not familiar with TLR4 structure and signaling (e.g. clinical hematologists) to have this description easily available in the article. However, these parts have been shortened and we have also reduced the number of references. Sections 2.1 and 2.2 have been shortened from 475 to 268 words, sections 2.3 and 2.4 are now included in the single section 2.3 and the overall length of these parts has been shortened from 315 to 250 words, and the original Table 1 is now included in the Supplementary information as Table S2.

What do you mean by preleukemic MDS? Yes, we know that MDS can go to AML.

Response: We would emphasize that this is only briefly mentioned and few references are included. However, in our opinion it is important to present clinical evidence that the effect of TLR4 is important at different steps of leukemogenesis (see section 2.2). We have therefore included this section also in our Revised Version, but the section is now shortened and the conclusion has been rewritten.

Overall I do not understand the reason for making this review. It is overly focused to the point that it seems like a research question.
I do not find a reason for this review to be published.

Response: It is very difficult for us to respond to this general statement when we revise our article because it is not followed by specific suggestions. There seems to be a disagreement with the other three reviewers that give a higher rating and consider our article as “very interesting, well writing and really rich in information”, “it addresses an emerging topic that involves a broad scope of basic research with promising translational aspects”, “very comprehensive” and  “very interesting”.

We have tried to include some additional comments that support the possible importance of TLR4 in AML, including a new section with recent references on the importance of IL1 in AML. A main signaling pathway used both by IL1R and TLR4 is the MyD88 mediator; in our opinion the common intracellular pathways for IL1R and TLR further support the hypothesis that the role of TLR4 in AML should be further investigated (section 8.2). Eight new references have been added as a part of this new chapter. We have also rewritten the last part of Section 1 introduction and Section 9 Conclusion. We hope the reviewer agrees that this has improved our review and justifies publication despite the initial comments made by this particular reviewer.

Reviewer 3 Report

Bruserud and colleagues revised in this manuscript the role of TLR4 in acute myeloid leukemia and bone marrow stroma cells as osteoblasts and osteoclasts. TLR4 is an important molecule in the immune system by its interaction with LPS and activation of the NFkB pathway. However, in the last years, a role in myeloid malignancies as AML or MDS has been described. Hence, it also affects proliferation and differentiation bone marrow cells as osteoblasts and osteoclasts, which also support the growth of leukemia cells.

Although the review of Bruserud et al is very comprehensive, some issues should be addressed prior publication.

Major points:

  • From my point of view, the review is too long. The authors described in the text with too many details, which are again described in the tables. The citation of 140 references are also too many. The authors should select, which ones are the most important for the work.

Minor points

  • There is a question mark in “even the megakaryocytes function as a kind (?) of stro-41 mal cells because they are involve” (page 1 line 40). This should be deleted

  • Point 4 “TLR4 in preleukemic MDS”: I would recommend change the title. MDS may be seen as a “pre”-leukemia but not all the patients transform to AML. Hence, most of the patients do not transform to AML. I would suggest just to say “TLR4 in MDS”.

The sentences in this paragraph are somehow very short and do not connect to each other. I would suggest to rewrite the paragraph connecting the several concepts of the paragraph.

  • there is a question mark in “and (?) ascorbic acid which will 352 increase osteoblastic differentiation” (see page 9, line 352). This should be deleted.

  • Point 3. “The TLR4 induced increase in extracellular release of soluble mediators by osteoblasts” (Page 9, line 370). Same comment as for point 4. The sentences in this paragraph are somehow very short and do not connect to each other. I would suggest to re-write the paragraph connecting the several concepts of the paragraph.

  • The text of the tables are centered, which makes a very confused impression. I would suggest to align the text to the left with or without justification.

Author Response

See enclosed letter.

Reviewer 4 Report

Molecules 1480617

Toll-like Receptor 4, Osteoblasts and Leukemogenesis; the Lesson from Acute Myeloid Leukemia

The manuscript is very interesting; however there are some points that the authors need to clear.

In the introduction they must describe the new in this review. Each section is interesting but such as this was described it is difficult to identify new points. On top of that, they must mention the date of their review, because in references they included less of 20% of references from 2020 to 2021. It is possible that these are all that exist, but they need to mention this.

In the table 1 and 2 they must search the way of simplifying them.

In many points they include between parentheses a symbol (?); it is not clear if they mention this as a hypothesis or which is the reason? They must mention the level of evidence about these points, this may be clearer.

In different parts of their manuscript the authors say that TLR4 is related with leukemogenesis, they should clarify if this is with all the leukemias or if it is only in the acute myeloid leukemia. This is repeated with relation to other molecules, they should clarify if specifically is with AML or with all the leukemias in general.

They may mention the clinical transcendence of TLR4 in future treatments again AML or its complications, such as the infections. They should highlight the questions that in new research should be asked.

Author Response

The manuscript is very interesting; however there are some points that the authors need to clear.

Response: We are very grateful for this general comment.

In the introduction they must describe the new in this review. Each section is interesting but such as this was described it is difficult to identify new points. On top of that, they must mention the date of their review, because in references they included less of 20% of references from 2020 to 2021. It is possible that these are all that exist, but they need to mention this.

Response: A PubMed search on TLR4 combined with acute myeloid leukemia resulted in 40 hits, only 9 of the identified references were from the 2019/20/21. The total number of publications was 32. When combining acute myeloid leukemia with toll-like receptor the total number of identified publications were 128, and 41 of them were published in 2019/20/21. Thus, even though we regard TLR to be of potential biological and clinical importance in AML and therefore deserving a review, relatively few new references are available.

When searching for Osteoblast plus toll like receptor 4 we identified 90 articles, but only 5 of them were published during the last 3 years. However, for the term osteoblast combined with toll-like receptor 4 we identified 197 articles, 52 of them being from the 2018/19/20/21. Thus, the interest for this field seems to be increasing.

Even though one may say that the publications are few, in our opinion the toll-like receptors are important in AML and a review is thereby justified. First, even though the studies are few they suggest an association with prognosis/survival/chemosensitivity. Second, these receptors are important in the crosstalk between cells in the bone marrow microenvironment. Finally, the downstream signaling (e.g. NFκB activation) and the crosstalk with other intracellular pathways (e.g. IL1-MyD88 signaling) all suggest that these receptors are important.

In the table 1 and 2 they must search the way of simplifying them.

Response: Table 1 is now Table S2 (see comments by reviewers 2 and 3); it has been redesigned and is hopefully easier to read. This table is included as background information for the reader and also as an illustration of the complexity of TLR4 ligation and downstream signaling.

The original Table 2 (now Table 1 in the Revised Version) has also been redesigned as suggested by reviewer 3 (see above).

In many points they include between parentheses a symbol (?); it is not clear if they mention this as a hypothesis or which is the reason? They must mention the level of evidence about these points, this may be clearer.

Response: We have carefully gone through these symbols and in most cases we could simply remove these question marks. The question marks in lines 41, 166, 352 and 439 of the original version could be removed, the section including the question mark on line 592 in the original version has been rewritten (page 12).

In different parts of their manuscript the authors say that TLR4 is related with leukemogenesis, they should clarify if this is with all the leukemias or if it is only in the acute myeloid leukemia. This is repeated with relation to other molecules, they should clarify if specifically is with AML or with all the leukemias in general.

Response: We have now corrected the text and refer to leukemogenesis in AML throughout the text. These alterations are marked with yellow (see pages 5,9 and 10).

They may mention the clinical transcendence of TLR4 in future treatments again AML or its complications, such as the infections. They should highlight the questions that in new research should be asked.

Response: We have added more detailed comments about future research in Section 9 conclusion.

Round 2

Reviewer 2 Report

It is better now.

Reviewer 4 Report

The manuscript is very interesting. I agree with the comments of the authors